# Simulation and Performance Comparison for $CO_2$ Capture by Aqueous Solvents of *N*-(2-Hydroxyethyl) Piperazine and Another Five Single Amines

Simeng Li, Han Li, Yanmei Yu and Jian Chen *

State Key Laboratory of Chemical Engineering, Tsinghua University, Beijing 100084, China; li-sm17@mails.tsinghua.edu.cn (S.L.); lihan0624@gmail.com (H.L.); yuyanm@mail.tsinghua.edu.cn (Y.Y.)
* Correspondence: cj-dce@mail.tsinghua.edu.cn

**Abstract:** *N*-(2-Hydroxyethyl) piperazine (HEPZ) has a chemical structure similar to PZ and has less volatility. It is not easy to volatilize in a continuous operation device. It is studied to replace PZ as a promotor to increase the $CO_2$ capture rate. This paper researches the lowest energy consumption and absorbent loss of $HEPZ/H_2O$ in the absorption-regeneration process, and compares it with another five amines, including PZ, MEA, 1-MPZ, AMP and DMEA. Based on the thermodynamic model, this work establishes a process simulation based on the equilibrium stage, assuming that all stages of the absorption and desorption towers reach thermodynamic equilibrium and $CO_2$ recovery in the absorption tower is 90%. By optimizing the process parameters, the lowest thermodynamic energy consumption and absorbent loss of process operation are obtained. Our results show that HEPZ as a promotor to replace PZ and MEA has significant economic value. The lowest reboiler energy consumption of HEPZ with the optimal process parameters is 3.018 $GJ/tCO_2$, which is about 35.2% lower than that of PZ and about 11.6% lower than that of MEA, and HEPZ has the lowest solvent loss. The cyclic capacity is 64.7% higher than PZ and 21.6% lower than primary amine MEA.

**Keywords:** $CO_2$ capture; absorption; *N*-(2-Hydroxyethyl) piperazine; reboiler heat duty; regeneration energy





## 1. Introduction

The problem of global warming caused by excessive emission of greenhouse gases continues to pose a severe threat to the natural environment. $CO_2$ is the main greenhouse gas [1]. The suppression of global warming and the reduction of $CO_2$ emissions is a concern for many countries. Carbon capture, utilization and storage (CCUS) is one of the required emission reduction methods that can reduce carbon emissions on a large scale in the short to medium term, including the capture, transportation and storage of $CO_2$. It is estimated that CCUS will contribute about 17% of the total cumulative emission reduction by 2035 [2]. There are three main types of $CO_2$ capture technologies: oxy-fuel combustion, pre-combustion capture and post-combustion capture. Compared with the former two, the post-combustion capture technology has the slightest modification to coal-fired power plants, and the lowest cost of transformation and the operation of the power plant. The changes are minimal, so it is suitable for upgrading existing power plants [3]. Among the methods for capturing $CO_2$ from flue gas of coal-fired power plants, amine scrubbing has been successfully applied in ammonia production and natural gas processes [4]. However, the regeneration of the absorbent is carried out by high-temperature heating, so the capture of $CO_2$ based on alcohol amine solutions requires huge regeneration energy. Typically, the energy penalty associated with solvent regeneration is the largest contributor to operating costs [3]. This is the biggest challenge to post-combustion capture still in need of a solution. The power generation efficiency of a typical 500 $MW_e$ supercritical power plant is about 44%, and when the carbon capture process is added, the power generation efficiency can

be reduced by 9.4–10.6% [5], and some power plants' power generation efficiency can lose up to 25% [3]. In order to reduce the cost, research has been carried out on several aspects: the development of new absorbents, high-efficiency separation equipment and enhancing technological processes.

The loss of organic amines in the absorption process roughly comes from the following scenarios: reactions with residual oxygen in the flue gas (oxidative degradation), thermal degradation at high temperatures (mainly occurs during the operation of the desorption tower) and volatilization of absorbents. Therefore, as an absorbent, amine has good $CO_2$ absorption and desorption capabilities. Still, it also needs to have less oxidative degradation and high-temperature degradation, along with low vaporization. The influence of the organic amine solution's viscosity cannot be ignored: the higher the viscosity, the slower the mass and heat transfer rate, which leads to an increase in the size of heat exchangers. Similarly, with a lower mass transfer and reaction rate, the need to increase the size of the absorption and desorption tower will also put pressure on the capital. Ideally, high-performance chemical absorbents generally need to have the following properties [6]: The large circulating absorption capacity can reduce the circulating volume of the absorbent, thereby reducing the power of the pump and reducing the energy consumption of the reboiler. The large absorption rate reduces the size of the absorption tower, thereby reducing facility investment. Low degradation and volatility can decrease the solvent replenishment and the environmental pollution, and low viscosity can reduce the size of packing, and of heat exchangers. The low heat of the reaction can reduce the energy consumption required for solvent regeneration.

Different organic amines have different structures, so their absorption characteristics are different due to various absorption mechanisms. Primary and secondary amines have a fast reaction rate, low absorption capacity and high absorption heat [7]. Tertiary amines and sterically hindered amines have high circulating absorption capacity, slow absorption rate and low heat of absorption [6,8,9]. The mixed amine system refers to a mixed system of amines with different reaction mechanisms, which combines the advantages of two alcohol amines with a high circulating absorption capacity and a rapid absorption rate. For example, the primary amine MEA and cyclic amine PZ with a fast absorption rate are added to a sterically hindered amine with a high capacity [10–12]. MEA is the most commonly used absorbent in the process of absorbing $CO_2$ by amine scrubbing, with the most extended history of use. Due to the accumulated process experience and performance, it is often considered as the reference solvent for new absorbents. Nevertheless, the regeneration energy of the MEA solvent to capture $CO_2$ is too high and MEA will degrade during desorption operation with high temperature, which will lead to an increase in the cost of overall carbon capture and is unacceptable for long-time, large-scale carbon capture deployment. Due to the fast absorption rate, PZ is usually added to the amine system as a promotor to increase the carbon absorption rate of the system. The disadvantage of piperazine as a promotor is its low boiling point and high melting point. It is easy to crystallize at low temperatures and cannot be configured with higher concentration solutions, therefore reducing the absorption capacity. Its boiling point of 146 °C is within 120–160 °C, the range of maximum working temperature of device, so it is easy to volatilize in the continuous absorption and desorption device and causes solvent loss. Then, PZ derivatives have similar molecular structures as PZ and have also been researched for their activation performance in recent years [13–15].

As one of the PZ derivatives, HEPZ has the highest boiling point of 246 °C and better thermal stability than PZ. Therefore, in industrial applications, it can withstand higher temperatures and potentially replace PZ as a promotor. However, only a part of the kinetic-related research [13,14] and $CO_2$ solubility of HEPZ/$H_2O$ has been studied [13]. There is a lack of thermodynamic modeling and process simulation of the system HEPZ/$H_2O$/$CO_2$. The advantage of HEPZ instead of PZ to capture $CO_2$ is only based on theoretical analysis of physical properties and has not been verified by experiments or process simulations. The pilot-scale study on the new absorbent can obtain its economic performance and process

energy consumption to evaluate and promote its industrial application. However, the pilot plant experiment requires time and expense. The new absorbent can be simulated in the Aspen Plus® platform before the pilot-scale study, and compared with the widely used absorbents such as MEA and PZ, in the energy consumption, absorbent loss and cyclic absorption capacity, to evaluate whether it has the value of the pilot-scale study.

With process simulation, parametric studies can be carried out for analysis of a post-combustion $CO_2$ capture plant. Abu-Zahra et al. [16] established a $CO_2$ capture process based on absorption/desorption with MEA solutions and conducted a parametric study of the economic performance. The optimization included investigating the effect of $CO_2$ removal percentage, MEA concentration, lean solvent loading and stripper operating pressure. In addition to the process design parameters, the impact of economic parameters such as fuel prices and interest rate was investigated. Liang et al. [17] presented the simulation of a monoethanolamine (MEA)-based $CO_2$ capture and compression process, and conducted an optimization of some important process parameters, including the operating stripper pressure, $CO_2$ capture efficiency and steam extraction location. Dash et al. [18] used the absorption-regeneration process with aqueous (AMP + PZ) solvents of 30–50 wt.% total amine concentration simulated by the RadFrac-RateSep block in the Aspen Plus® platform to study the effect of amine concentration, L/G, column pressure and packing height on $CO_2$ capture rate, the effect of column pressure on temperature profile and the reboiler duty for $CO_2$ capture at a constant L/G. Dubois and Thomas [19] studied different $CO_2$ capture process configurations applied to the flue gas coming from the cement plant and using three different solvents: MEA, PZ and PZ-MDEA blend. For each configuration and solvent, parametric studies were carried out in order to identify the operating conditions (L/G, split fraction, flash pressure variation, etc.) minimizing the solvent regeneration energy. Ayittey et al. [20] carried out detailed parametric analyses on a post-combustion $CO_2$ capture system using a hot potassium carbonate solution using a rate-based simulation study. The effect of system parameters on $CO_2$ capture rate and stripper reboiler duty were studied: reflux ratio of stripping column, lean solvent flowrate, flue gas flowrate, lean solvent concentration, lean solvent temperature, flue gas temperature and absorber operating pressure. Changes in the lean solvent concentration (from 25 to 45 wt.%) and absorber operating pressure (from 0.5 to 2.5 MPa) were observed to have the most significant effects on the overall performance of the capture system.

Our previous research measured the thermodynamic properties and established a $HEPZ/H_2O/CO_2$ model in Aspen Plus® platform. The physical parameter system required to establish a process simulation has been obtained. In this work, a process simulation model was established based on the previous thermodynamic model to study the lowest energy consumption and corresponding operating parameters in the operation of the process. The operating parameters of the process with the lowest energy consumption were explored and the cyclic capacity and absorbent loss of HEPZ, PZ and MEA under the lowest energy consumption were compared.

## 2. Materials and Methods

### 2.1. Absorbent

The abbreviations and molecular structures of the absorption reagents studied in this work are shown in Figure 1. Table 1 lists some of the physical property parameters and information of these solvents. In addition to the new absorbent HEPZ that is mainly studied, there are also a variety of absorbents used for comparative evaluation of its performance. According to molecular structures, the amine absorbents currently studied in the literature are classified into primary amines, secondary amines, tertiary amines, sterically hindered amines and polyamines. In order to thoroughly compare the performance difference between HEPZ and different absorbents, the primary amines: monoethanolamine (MEA), piperazine (PZ) and piperazine derivative 1-methyl piperazine (1-MPZ), sterically hindered amine 2-amino-2-methyl-1-propanol (AMP) and tertiary amine N, *N*-dimethylethanolamine (DMEA), were selected and studied. MEA has the

longest industrial application and is usually used as a benchmark for comparison between various absorbents [21–24]. Concerning, 1-MPZ, with a molecular structure similar to PZ, its solubility and absorption rate have been shown by many studies [13,14,25,26]. AMP is a sterically hindered amine, and its product reacted with $CO_2$ is unstable and easy to desorb, which is the same as DMEA—the isomer of AMP. Both have low absorption heat and desorption heat, but their absorption rate is slow [27], and the $CO_2$ solubility at low temperature and low pressure is poor [28]. The thermodynamic parameters of the above absorbents were all obtained by the method in Section 2.2.2.

**Figure 1.** Molecular structure of studied amines.

**Table 1.** The physical parameters and information of solvents.

| Absorbent | CAS# | BP/°C | MW |
|---|---|---|---|
| MEA | 141-43-5 | 170.8 | 61.08 |
| PZ | 110-85-0 | 146–148 | 86.14 |
| 1-MPZ | 109-01-3 | 138 | 100.16 |
| DMEA | 108-01-0 | 133–136 | 89.14 |
| AMP | 124-68-5 | 165 | 89.14 |
| HEPZ | 103-76-4 | 246 | 130.19 |

*2.2. Process Simulation*

2.2.1. The Modeling Object

In the process of amine scrubbing, pumps, condensers, reboilers and compressors all consume energy. The two most energy-consuming parts are the reboiler and compressor. The compressor is used to compress the high-purity $CO_2$ gas desorbed from the top of the regenerator. Therefore, its energy consumption depends only on the pressure of the regenerator. If the pressure of the regenerator is high, the compression energy consumption will be low, while the pressure of the regenerator depends on the high resistance to oxidative and thermal degradation of the absorbent. This is due to the fact that high desorption pressure will make the desorption temperature higher. Different absorbents have different resistance to degradation at high temperatures, and absorbents that can withstand higher temperatures can be desorbed at higher pressures. The degradation of absorbents is not within the scope of this work, and all absorbents are compared under the same desorption pressure, so their compression energy consumption is the same. The energy consumption of the reboiler is determined by the $CO_2$ cyclic capacity and the heat of $CO_2$ desorption of the absorbent, which is the main research content in this work. Firstly, an accurate thermodynamic model of the capture system was established using data from experiments and the literature. On this basis, a process simulation based on the thermodynamic-equilibrium model was established, and it was assumed that both the absorber and the regenerator reached thermodynamic equilibrium and the influence of kinetics was not considered. The lowest energy consumption of each absorbent was obtained by optimizing

the process parameters. In order to evaluate the performance of HEPZ instead of PZ as a promotor, simulation models of HEPZ, PZ and MEA as absorbent systems were established in this work, and the energy consumption, cyclic absorption and absorbent loss of the three were compared, respectively.

### 2.2.2. Physical Parameter System

In the processing system, there are components $H_2O$, $CO_2$, $N_2$, HEPZ, $H_3O^+$, $OH^-$, $HCO_3^-$, $CO_3^{2-}$, $HEPZH^+$, $HEPZH_2^{2+}$, HEPZCOOH and $HEPZCOO^-$, and the physical property method is ENRTL-RK. Establishing a process simulation model requires a complete physical parameter system, including the necessary physical parameters of amines in the system, the interaction parameters of the nonrandom two-liquid model (NRTL) and the electrolyte nonrandom two-liquid (ENRTL) activity coefficient model, along with the standard state properties of amine ions. The physical property parameters and interaction parameters between the HEPZ molecule and various ions in the system are lacking in the Aspen database. It is necessary to perform regression fitting on the heat capacity, saturated vapor pressure and vapor–liquid balance and solubility of amine data. A rigorous thermodynamic model was established to obtain the physical property system, and the specific modeling method is described in detail in [29]. In our previous work, $CO_2$ solubility was measured for HEPZ aqueous solutions at three concentrations—5, 15 and 30 wt.%, and four temperatures—313.15, 343.15, 373.15 and 393.15 K. The VLE data for $HEPZ/H_2O$ were obtained at a pressure of 30–100 pKa, within a whole mole-fraction range. By regressing and fitting data from experiments and the literature, the parameters of NRTL and ENRTL were obtained, and the standard thermodynamic parameters of the new substances $HEPZH^+$ and $HEPZH_2^{2+}$, which were not in the database of Aspen, were manually adjusted to fit the dissociation constant of HEPZ. The equilibrium constant can be calculated from the standard state properties of the reactants and products. Knowing the standard Gibbs free energy, the standard enthalpy of each component's formation and the heat capacity in the reference state in the reaction equilibrium equation, the equilibrium constant of each reaction can be calculated. In the work of [30], we could obtain pKa1 and pKa2 of HEPZ, so the equilibrium constants of the hydrolysis reactions of HEPZ could also be obtained. Therefore, the calculated equilibrium constant can be used to verify whether the standard state properties of the reactants and products are correct. The pKa curve was compared with the experimental data in [30], and the value of the standard property is manually adjusted if there is any deviation. In this work, the obtained pKa curve agreed well with the experimental data. More detailed descriptions can be found in [31].

This accurate thermodynamic model can predict the thermodynamic properties and absorption performance of the $CO_2$ capture system, which provides reliable physical properties for the establishment of process simulation. The physical parameters of the model are listed in the previous work [31].

### 2.2.3. Process Description and Main Process Parameter Setting

A schematic diagram of the amine scrubbing process simulation is shown in Figure 2. The main equipment in the process is an absorber and regenerator, and the auxiliary equipment includes pumps, heat exchangers, condensers and flash tanks. The absorber and regenerator are modeled by RadFrac in Aspen Plus® platform, using the equilibrium model. FLUEGAS with a temperature of 110 °C, a pressure of 1 bar and a flow rate of 499.8 kg/h is first flashed by a flash evaporator, the purpose of which is to reduce the flue gas temperature to 40 °C and ensure that the water in the flue gas reaches saturation. The composition of flue gas is (volume fraction): 8% $H_2O$, 10% $CO_2$, 76% $N_2$ and 6% $O_2$, which was referenced by the experimental values of the pilot plant. Then, GASIN enters from the bottom of the absorber, contacts with the absorbent in the countercurrent and is evacuated from the top of the absorber after condensation, separation and liquid reflux. The temperature of the condenser is 40 °C, and the temperature of LS1 is also set to 40 °C. After RICHOUT is pressurized to 2.5 bar by the pump, the HS1 exchanges heat

with the high-temperature lean liquid from the regenerator. The lean-rich heat exchanger is modeled using the Aspen® Heatx model. The minimum heat transfer temperature difference is set to 10 °C, and the temperature of the condenser at the top of the desorption tower is set to 40 °C. The number of plates in the absorber is sufficient to ensure that the absorber reaches the thermodynamic equilibrium, which is generally 40. When operating the process, the converged calculation result of the operation for the absorption tower with 40 plates was basically the same as the result for the absorption tower with more plates. For the desorption tower, the thermodynamic equilibrium can be achieved with fewer plates by our simulation, and it is sufficient to set the number of plates in the regenerator to 10, as suggested by Oyenekan and Rochelle [32]. The top plate's pressure of the absorber is 1 bar, and the pressure drop of 0.2 bar is considered for both towers. Some design specifications for the reboiler duty and the lean-rich heat exchanger are incorporated to converge the flow sheet in a closed loop. The design specifications require that the $CO_2$ removal ratio reaches 90% and the $CO_2$ loading of LS1 is equal to the $CO_2$ loading of LS2. The value range of the flow rate of LS1 and reboiler duty is set and the program is run. The program can calculate the accurate values of the flow rate of LS1 and reboiler duty. The parameters of LS1 with the lowest reboiler duty are explored by changing the amine concentration and $CO_2$ loading.

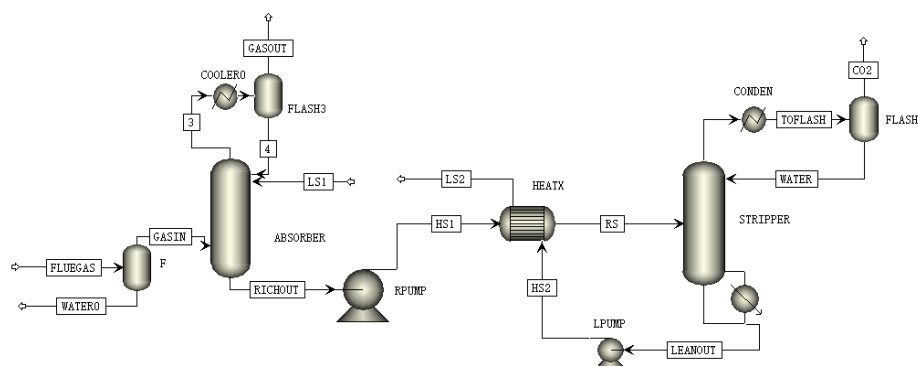

**Figure 2.** Schematic diagram of amine scrubbing process simulation.

*2.3. Regeneration Energy Study*

This work studied the regeneration energy consumption, absorbent loss and cyclic capacity of different absorbents in the process of capturing $CO_2$. The goal of optimizing process parameters is to obtain the lowest energy consumption of the reboiler at 90% $CO_2$ recovery. The adjustable process parameters include solvent concentration, lean loading of LS1 and liquid–gas ratio, L/G. When the reboiler heat duty reaches the minimum value, the cyclic capacity and absorbent loss of different absorbents for capturing $CO_2$ can be obtained. The absorbent with the lowest regeneration energy consumption, the least absorbent loss and the largest cyclic capacity has the best economic benefits and potential for factory promotion.

The regeneration energy is determined by the $CO_2$ cyclic capacity of the absorbent and the heat of $CO_2$ desorption, which is the main research content of this study. The energy consumption of the reboiler is mainly used in three parts: water evaporation, $CO_2$ desorption and liquid heating. The evaporation heat of water and the liquid heating heat are determined by the $CO_2$ cyclic capacity. The higher the $CO_2$ cycle absorption, the lower the amount of absorbent, which will reduce the water evaporation heat and heat for liquid heating. In order to analyze the total energy consumption, that is, the composition of the reboiler energy consumption, $Q_{reb}$, the desorption part, including the heat exchanger, can be calculated as follows:

$$Q_{reb} = H_{CO_2} + H_{leanin2} - H_{toheatx1} - Q_{con} \tag{1}$$

where $Q_{con}$ is the heat of condensation of water, $H_{CO_2} + H_{leanin2} - H_{toheatx1}$ is the sum of the heat $Q_T$ required to heat the rich liquid at the bottom outlet of the absorption tower and the heat $Q_{abs}$ required to desorb from the rich loading to the lean loading. $Q_T$ is obtained directly from the heat capacity, and the calculation formula is as follows:

$$Q_T = Cp_{toheatx1} \times F_{mass,toheatx1} \times (T_{leanin2} - T_{toheatx1} / F_{mass,CO_2}$$ (2)

where $Cp_{toheatx1}$ and $F_{mass,toheatx1}$ are the heat capacity and mass flow rate of rich liquid TOHEATX1, respectively, and $F_{mass,CO_2}$ is the mass flow rate of $CO_2$ at the top of the desorption tower. After obtaining $Q_T$, the value of $Q_{abs}$ can be calculated as:

$$Q_{abs} = Q_{reb} + Q_{con} - Q_T$$ (3)

This work studies the influence of absorbent concentration and process parameters on the total energy consumption of the reboiler and its three parts. By comparing energy consumption under the optimal operating conditions of each absorbent, the potential of HEPZ as a new absorbent is evaluated. In Section 3.1, the effect of these process parameters on the reboiler energy of each absorbent is discussed. According to the discussion, the optimal process parameters that could minimize the energy consumption of the reboiler were obtained, as presented in Section 3.2. When optimizing process parameters, the effect of a certain process parameter was studied while keeping other process parameters fixed. The optimal value of one process parameter was selected, then the optimal value of the next process parameter was studied.

## 3. Results and Discussion

### 3.1. Influence of Process Parameters on Reboiler Energy

#### 3.1.1. Solvent Concentration

When the $CO_2$ partial pressure of flue gas is 10 kPa and the pressure of the desorption tower is 2.2 bar, the influence of concentration on reboiler energy consumption is studied, and the studied concentration range is 10–40 wt.%. Since the measured concentration of the $CO_2$ solubility data of the HEPZ solution used for regression modeling was up to 30 wt.%, the highest solution concentration applicable to the HEPZ thermodynamic model is also 30 wt.%. The concentrations studied for HEPZ were 5, 15 and 30 wt.%. When the concentration of PZ is too high, crystals will be precipitated and the solubility of anhydrous PZ at 20 °C is 14 wt.% PZ, which corresponds to 1.9 m PZ [32]. Therefore, problems will exist during plant start-up, operation and shutdown. The relationship between its concentration and energy consumption will no longer be studied. Figure 3 shows the relationship between the concentration of MEA, HEPZ, DMEA, 1-MPZ, AMP aqueous solution and energy consumption.

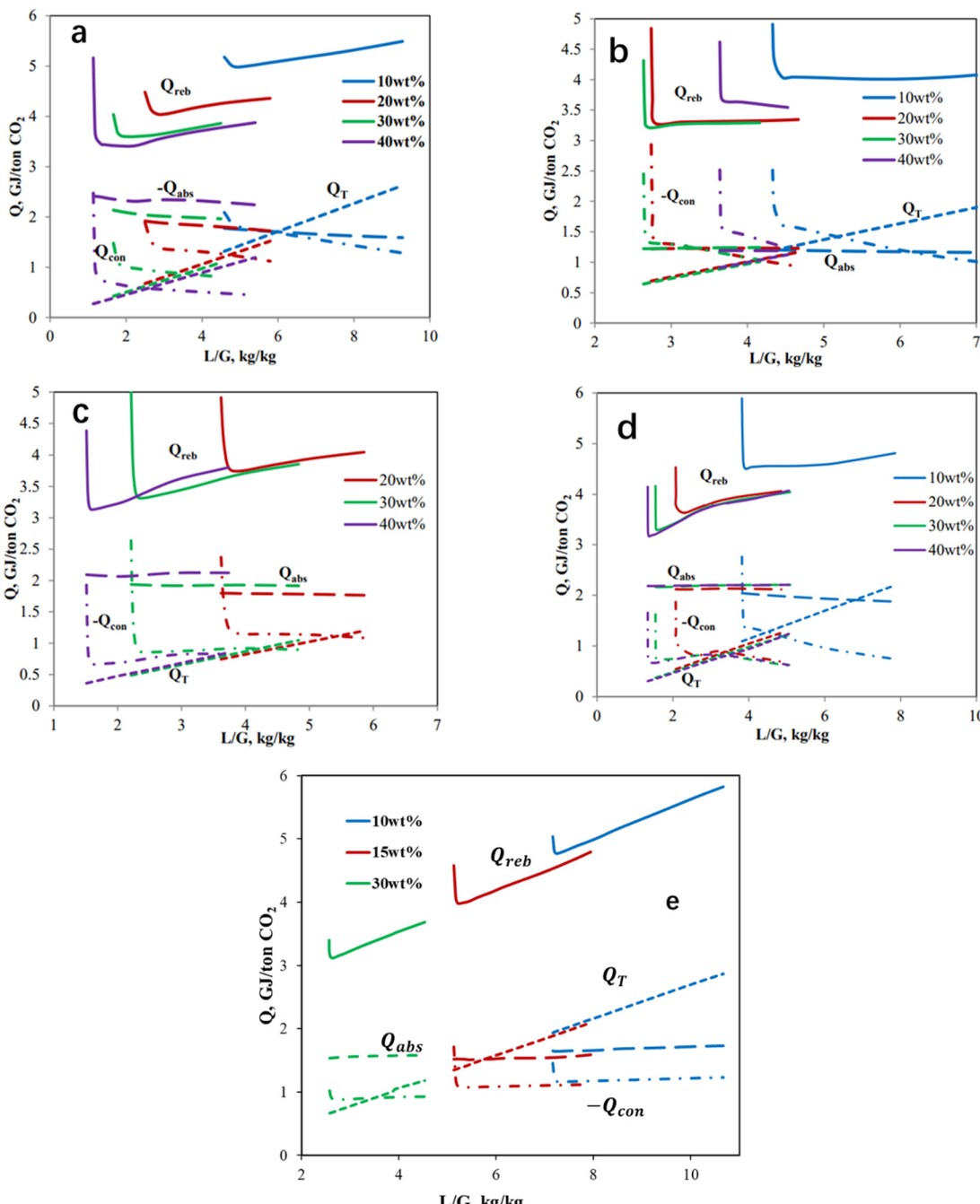

**Figure 3.** When the $CO_2$ partial pressure of flue gas is 10 kPa, and the pressure of the desorption tower is 2.2 bar, the influence of the concentration of aqueous amine solutions on energy consumption for: (**a**) MEA, (**b**) DMEA, (**c**) 1-MPZ, (**d**) AMP and (**e**) HEPZ, is shown.

It can be seen from Figure 3 that as the concentration of the MEA aqueous solution increases, $Q_{abs}$ increases and L/G of the absorption tower decreases, so $Q_{con}$ and $Q_T$ decrease. The sum of the three kinds of heat, $Q_{reb}$, decreases, and the magnitude of the decrease becomes smaller as the concentration increases. The concentration of DMEA aqueous solution increases from 10 to 30 wt.%, $Q_{abs}$ is basically unchanged, L/G decreases, so $Q_{con}$ and $Q_T$ decrease, and $Q_{reb}$ decreases. When it rises from 30 to 40 wt.%, L/G increases, $Q_{con}$ and $Q_T$ also increase, while $Q_{abs}$ is unchanged, so $Q_{reb}$ increases, and the optimal concentration of DMEA is 30 wt.%. As the concentration of 1-MPZ increases, $Q_{reb}$ decreases. The optimal concentration of 1-MPZ is 40 wt.%. The effect of the concentration

of AMP on energy consumption is the same as that of MEA. As for HEPZ, the reboiler energy consumption is significantly lower than that of other solvents, and the proportion of $Q_{con}$ is the lowest, which does not change much with the concentration. Due to the large MW of HEPZ, the L/G difference of HEPZ solutions with different concentrations is significant, and $Q_{con}$ and $Q_T$ decrease significantly with the increase of the concentration.

### 3.1.2. L/G

It can be seen in Figure 3 that L/G has different effects on reboiler energy consumption and its three parts. With the increase of L/G, $Q_{abs}$ is basically unchanged, and $Q_T$ increases linearly, while $Q_{con}$ first decreases and then increases. The sum of total energy consumption, $Q_{reb}$, first decreases and then increases with the increase of L/G. The stronger the amine's cyclic capacity, the smaller the L/G and the amount of solvent used. When L/G drops to a certain value, $Q_{reb}$ suddenly increases significantly, indicating that the cyclic capacity cannot increase indefinitely. The lowest energy consumption and relatively strong cyclic capacity can be obtained at a certain value, which is shown in Figure 4. In addition, it can be seen from Figure 4 that for different solvents, the reboiler energy consumption changes with the cyclic capacity in the same trend.

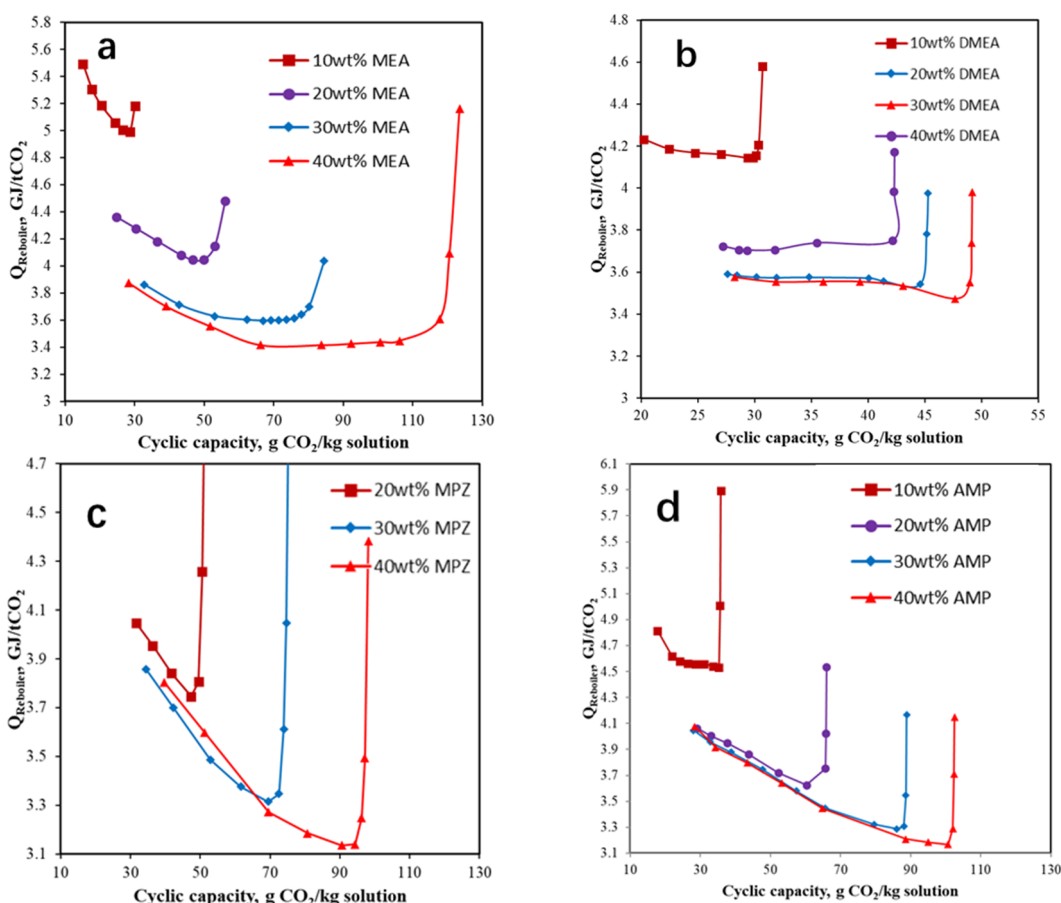

**Figure 4.** *Cont.*

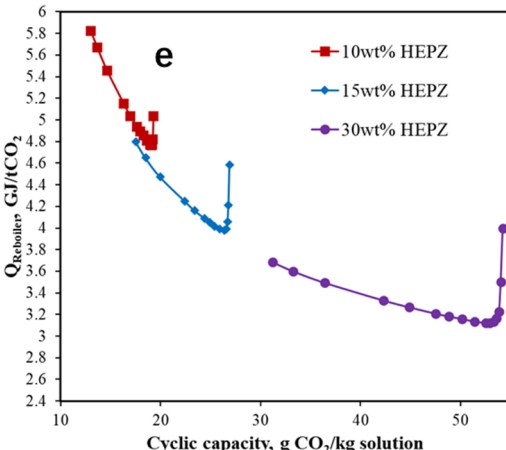

**Figure 4.** The relationship between reboiler energy consumption, $Q_{reb}$, and circulating absorption: (**a**) MEA, (**b**) DMEA, (**c**) 1-MPZ, (**d**) AMP and (**e**) HEPZ.

### 3.1.3. $CO_2$-Lean Loading

The previous section explained the influence of L/G on energy consumption, and the value of L/G is related to the cyclic capacity of the amine solvent. $CO_2$-lean loading, a process parameter set in the process, also affects the regeneration energy consumption by affecting the cyclic capacity. It is the $CO_2$ loading of the stream LS1. The influence relationship is shown in Figure 5. In the case of a fixed $CO_2$ removal rate, if the $CO_2$-lean loading is different, the rich loading is different. The cyclic capacity of the absorbent with different lean and rich loading is different. This can be seen from the $CO_2$ solubility curve of the absorbents. The cyclic capacity decreases as the $CO_2$-lean loading increases, which is the same for all solvents.

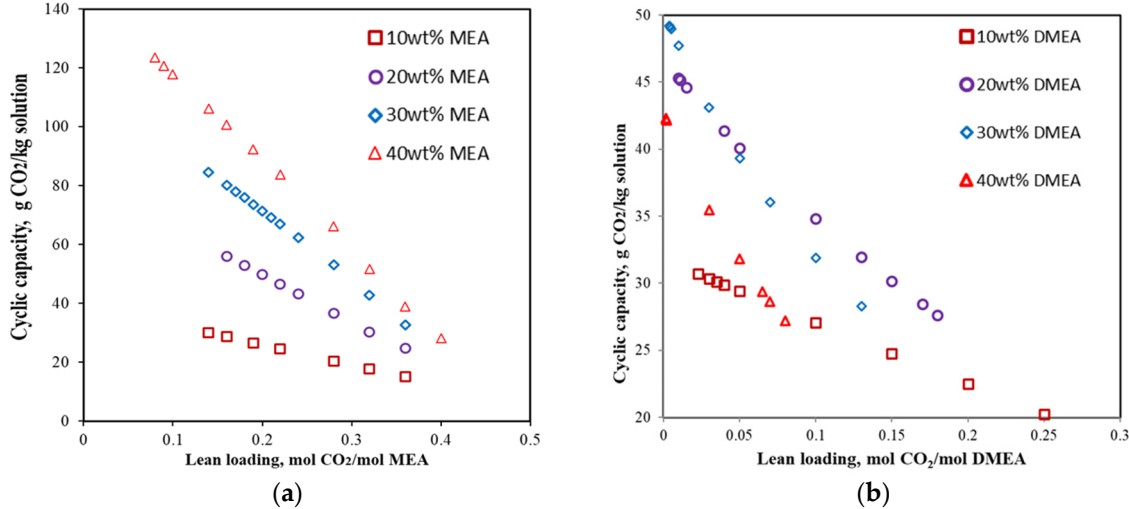

**Figure 5.** *Cont.*

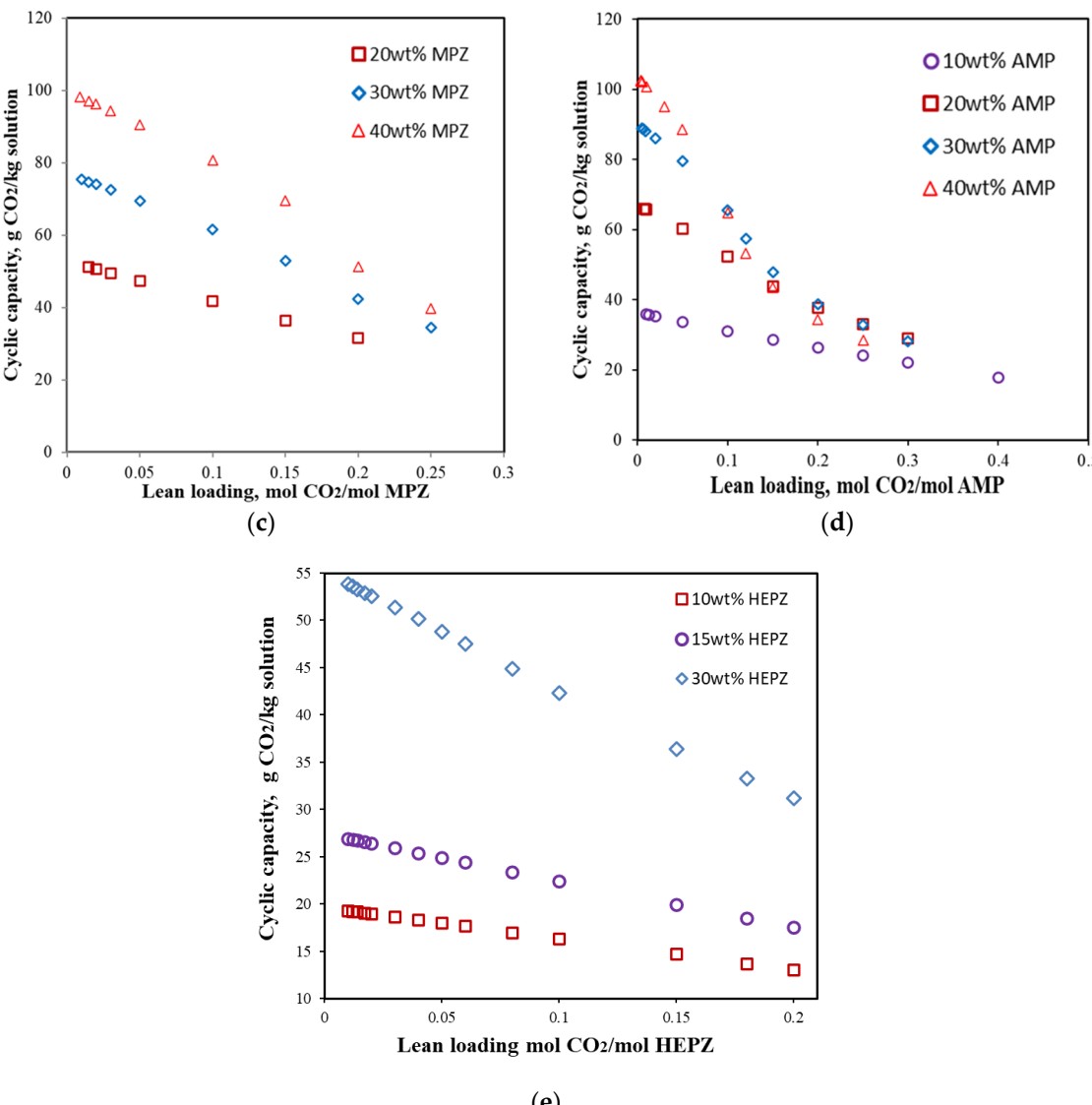

**Figure 5.** The influence relationship between the cyclic capacity and $CO_2$-lean loading: (**a**) MEA, (**b**) DMEA, (**c**) 1-MPZ, (**d**) AMP and (**e**) HEPZ.

### 3.1.4. $CO_2$ Partial Pressure of Flue Gas

The optimal concentration of different solvents with the lowest energy consumption was obtained in Section 3.1.1. The choice of PZ aqueous solution concentration was 10 wt.% because PZ was usually used as a promotor in mixed amine solutions. Although the concentration of $CO_2$-loaded PZ solution which will not crystallize can be higher [33], the use of a high-concentration PZ solution will still cause problems during plant start-up, operation and shutdown. In this work, the highest PZ concentration can be configured to be 10 wt.%, which was selected to compare with other absorbents. Figure 6 shows the $CO_2$ solubility curve of various solvents at 313.15 K. The curve slope shows the effect of $CO_2$ partial pressure on a load of carbon dioxide: the more significant the slope, the less the effect of the increase in $CO_2$ partial pressure on the $CO_2$ loading.

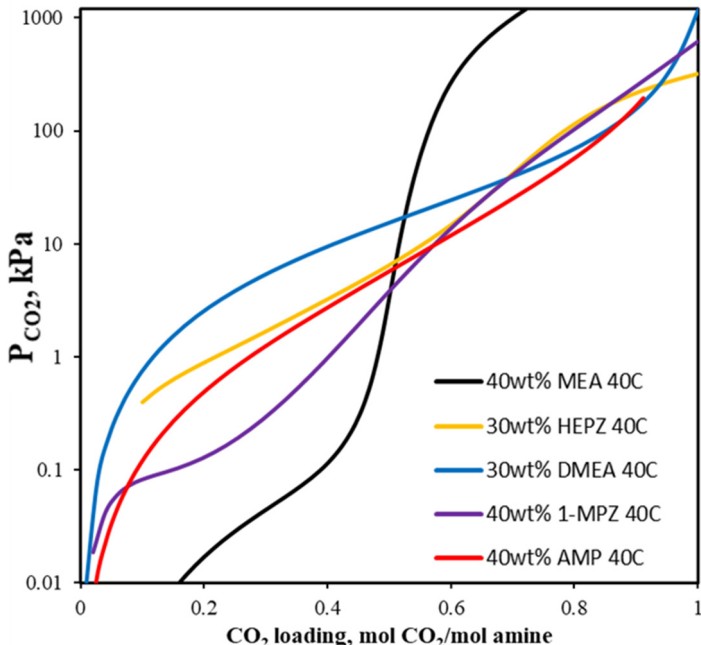

**Figure 6.** The $CO_2$ solubility curve of various solvents at 313.15 K.

The process simulation results shown in Figure 7 can verify the effect of $CO_2$ partial pressure on the regeneration energy consumption of different solvents. The solubility curve of primary amine MEA has the largest slope, which means that the change in $CO_2$ partial pressure has little effect on $CO_2$ loading. When the partial pressure is increased from 1 to 10 kPa, the $CO_2$ loading is almost unchanged. Therefore, it can be predicted that the increase of the $CO_2$ partial pressure of flue gas has no effect on the energy consumption of 40 wt.% MEA. When the partial pressure is increased from 5 to 15 kPa, the reboiler energy consumption is only reduced by about 0.1 GJ/t$CO_2$. The slope of the solubility curve of HEPZ and DMEA is the smallest, so the change of $CO_2$ partial pressure has a greater impact on $CO_2$ loading. The $CO_2$ partial pressure of flue gas was increased from 5 to 15 kPa, and the energy consumption of 30 wt.% HEPZ was reduced from 2.7 to 2.4 GJ/t$CO_2$, a reduction of approximately 11%. The energy consumption of 30 wt.% DMEA decreases more with the increase of the $CO_2$ partial pressure. When the partial pressure increases from 5 to 15 kPa, the energy consumption decreases from 3.8 to 3.05 GJ/t$CO_2$, a reduction of nearly 20%. For 40 wt.% 1-MPZ and 40 wt.% AMP, the slope of the solubility curve is between 40 wt.% MEA and 30 wt.% DMEA, so the reduction in energy consumption as $CO_2$ partial pressure increases is between 40 wt.% MEA and 30 wt.% DMEA. When the $CO_2$ partial pressure is increased from 5 to 10 kPa, the reboiler energy consumption of 40 wt.% 1-MPZ and 40 wt.% AMP are reduced by about 0.3 and 0.2 GJ/t$CO_2$, respectively. When the partial pressure is increased from 10 to 15 kPa, the energy consumption of 1-MPZ and AMP reboilers does not change significantly. The $CO_2$ partial pressure of flue gas mainly affects $Q_{con}$, and has almost no effect on $Q_{abs}$ and $Q_T$.

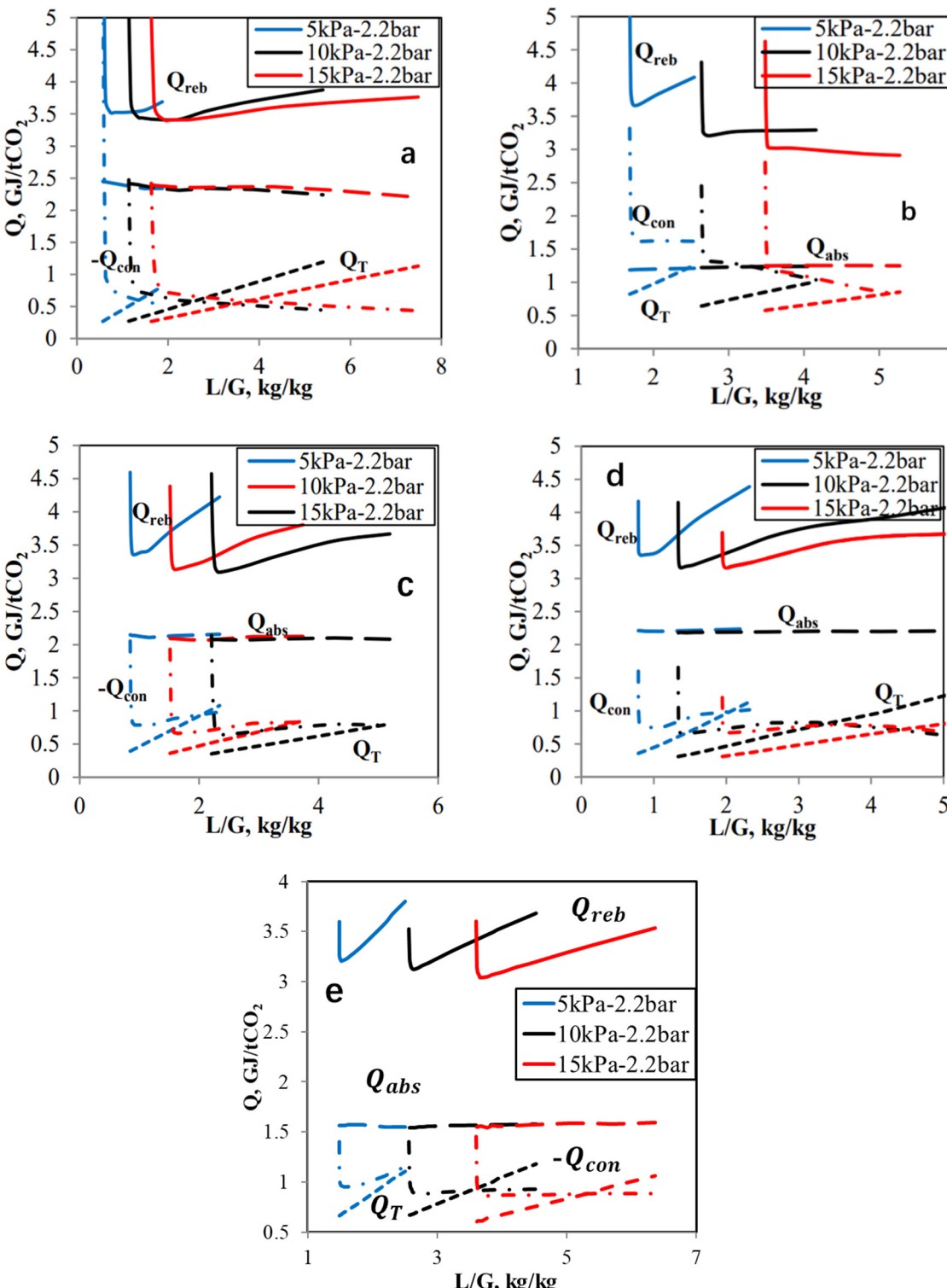

**Figure 7.** The effect of $CO_2$ partial pressure on the regeneration energy consumption of different solvents: (**a**) 40 wt.% MEA, (**b**) 30 wt.% DMEA, (**c**) 40 wt.% 1-MPZ, (**d**) 40 wt.% AMP and (**e**) 30 wt.% HEPZ.

### 3.1.5. Pressure of Desorption Tower

When other conditions remain unchanged, the change in the pressure of the desorption tower only affects $Q_{con}$, whereas $Q_T$ and $Q_{abs}$ are basically unaffected. As the pressure increases, $Q_{con}$ and $Q_{reb}$ decrease. Meanwhile, the increase of pressure will increase the temperature of the reboiler, and a high temperature will cause the degradation of

the absorbent. Figure 8 shows the effect of desorption tower pressure on the energy consumption of solvents and the reboiler temperature when the $CO_2$ partial pressure of flue gas is 10 kPa. The pressures of the desorption towers studied in this work are 1.2, 1.7 and 2.2 bar, respectively. The results of different solvents affected by the pressure of the desorption tower are consistent. For MEA, when the pressure is increased by 0.5 bar, the energy consumption, $Q_{reb}$, is reduced by about 0.1 GJ/tCO$_2$. For AMP and 1-MPZ, the energy consumption, $Q_{reb}$, decreases slightly as the pressure increases. For HEPZ, the energy consumption, $Q_{reb}$, decreases by approximately 0.2 GJ/tCO$_2$ as the pressure increases by 0.5 bar. For DMEA, when the pressure is increased from 1.2 to 1.7 bar, $Q_T$ used to heat the liquid is greatly reduced, and $Q_{reb}$ is reduced by 0.3 GJ/tCO$_2$. The minimum heat transfer temperature difference appears at the cold end of the heat exchanger with pressures of 1.7 and 2.2 bar; that is, the temperature difference between LS2 and HS1 in Figure 2 is 10 °C. However, when the pressure of the desorption tower is 1.2 bar, the minimum heat transfer temperature difference appears at the hot end of the heat exchanger, causing the temperature difference between LS2 and HS1 to be greater than 10 °C, so $Q_T$ is relatively higher. If the pressure of the desorption tower is increased by 0.5 bar, the energy consumption will be slightly reduced, and the desorption tower temperature will increase by approximately 10 °C. Since the maximum temperature of the solvent solubility measured in this work was 120 °C, the applicable temperature of the thermodynamic model was 120 °C. Therefore, the maximum pressure of the desorption tower is 2.2 bar.

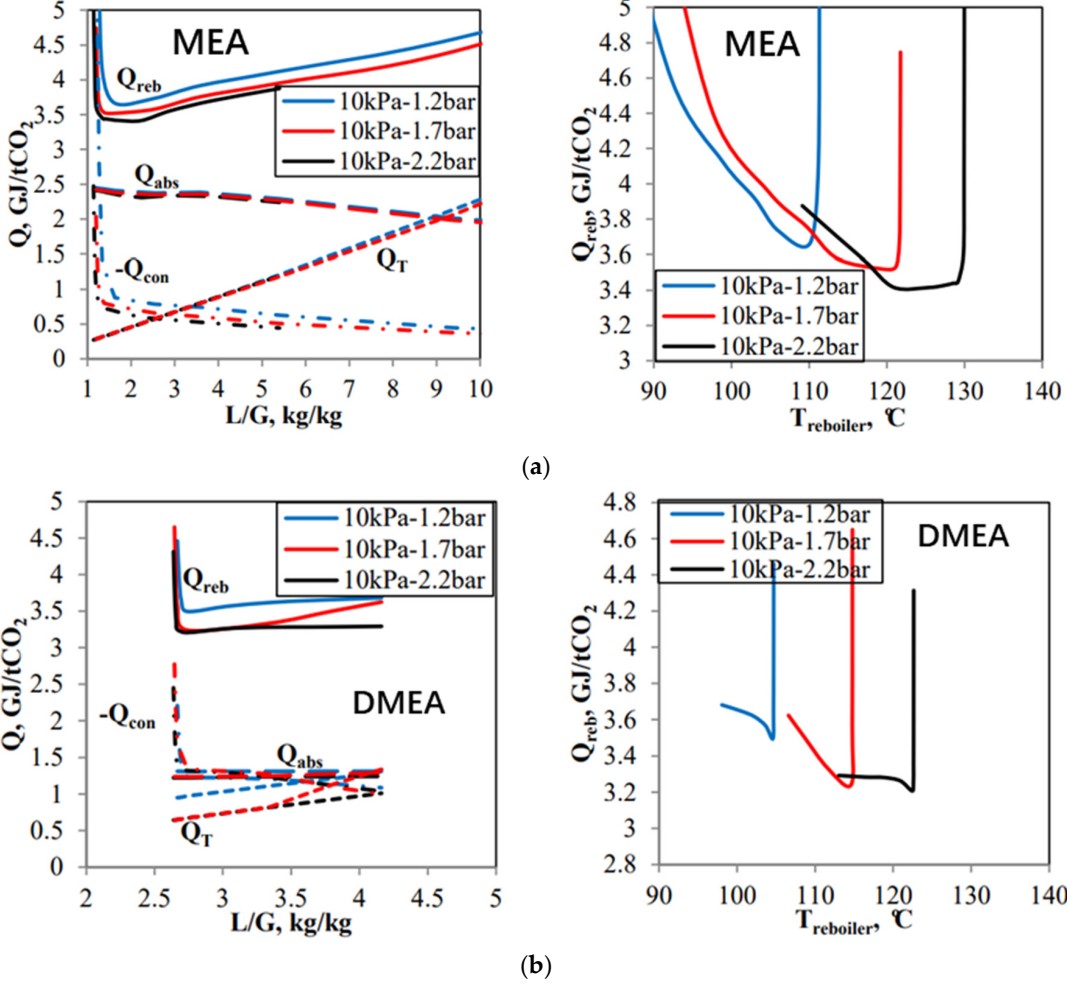

**Figure 8.** *Cont.*

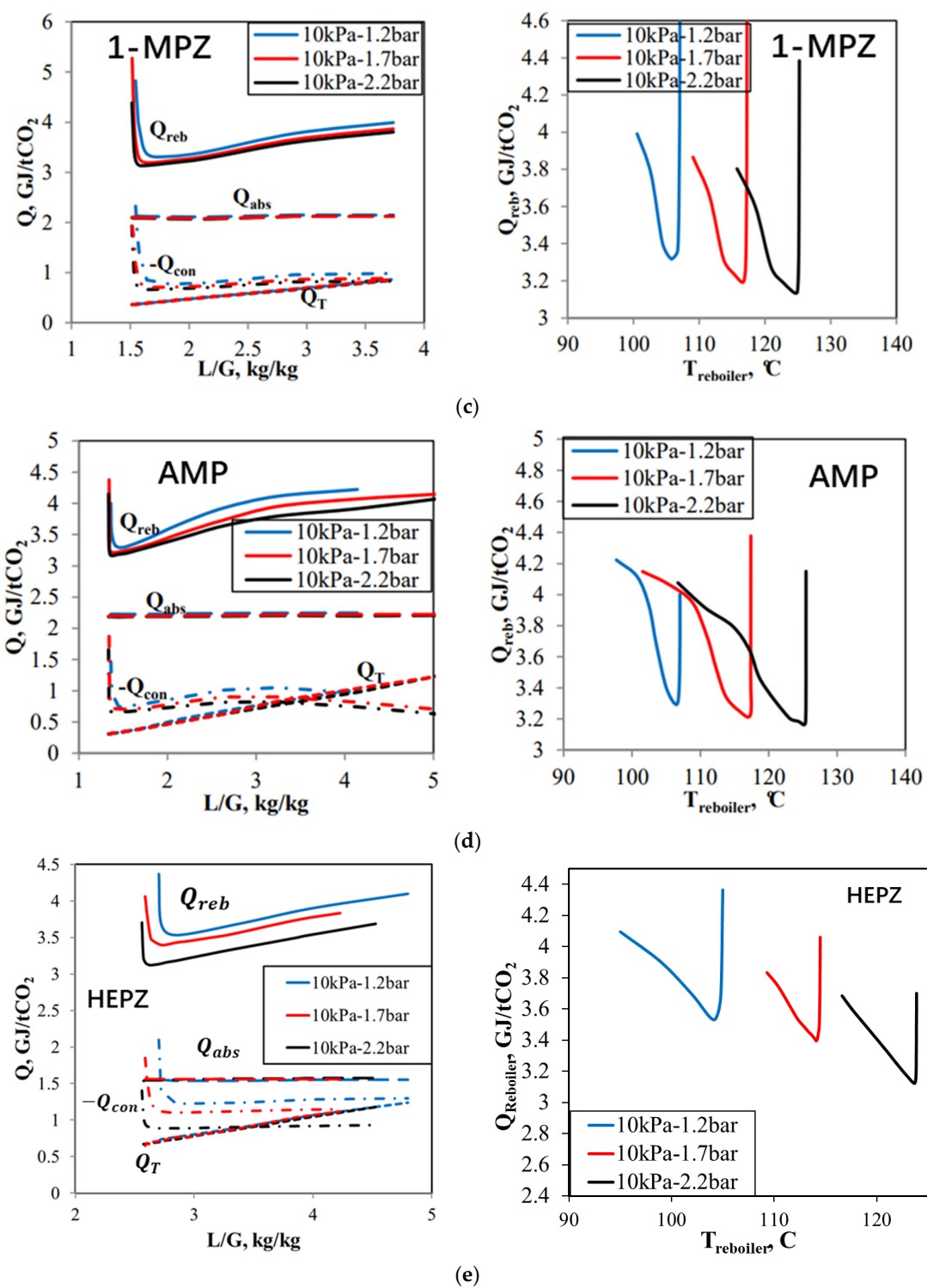

**Figure 8.** The effect of desorption tower pressure on reboiler energy consumption and reboiler temperature: (**a**) MEA, (**b**) DMEA, (**c**) 1-MPZ, (**d**) AMP and (**e**) HEPZ.

### 3.2. The Proportion of Different Heat in Reboiler Energy with the Optimal Process Parameters

From the results above, the optimal process parameters and the lowest reboiler energy for different solvents can be obtained, which are listed in Table 2. The concentration of each solvent is selected in Section 3.1, the $CO_2$ partial pressure of flue gas is selected as 10 kPa and the pressure of the desorption tower is 2.2 bar. The highest configurable concentration of PZ considered in this work, 10 wt.%, was selected as the concentration of the PZ solvent,

and the reason for this has been provided in Section 3.1. Due to the low concentration of PZ, L/G is 6.167, which is too large, and the reboiler energy consumption is very high, so PZ is not suitable for use as an absorbent alone. The L/G of other solvents is between 1.3 and 2.7, which is a reasonable range.

**Table 2.** The lowest reboiler energy and process parameters of different solvents.

| Absorbent | wt.% | $\alpha_{lean}$ mol/mol | $\alpha_{rich}$ mol/mol | L/G kg/kg | $T_{reb}$ °C | $Q_{reb}$ | $-Q_{con}$ | $Q_{abs}$ | $Q_T$ |
|---|---|---|---|---|---|---|---|---|---|
| | | | | | | GJ/ton $CO_2$ | | | |
| MEA | 40 | 0.22 | 0.511 | 1.742 | 125.4 | 3.415 | 0.669 | 2.347 | 0.401 |
| PZ | 10 | 0.20 | 0.846 | 6.167 | 123.3 | 4.657 | 2.008 | 1.604 | 1.05 |
| 1-MPZ | 40 | 0.05 | 0.565 | 1.641 | 124.5 | 3.136 | 0.662 | 2.084 | 0.392 |
| DMEA | 30 | 0.01 | 0.349 | 2.738 | 122.4 | 3.207 | 1.317 | 1.222 | 0.669 |
| AMP | 40 | 0.01 | 0.520 | 1.361 | 125.3 | 3.168 | 0.67 | 2.184 | 0.315 |
| HEPZ | 30 | 0.02 | 0.538 | 2.641 | 123.7 | 3.018 | 0.854 | 1.544 | 0.686 |

Figure 9 shows the lowest reboiler energy consumption of different solvents and the contribution of the three types of heat for different solvents at the optimal concentration. The highest reboiler energy consumption is PZ, because L/G is relatively high, so the energy consumption for condensing and heating the liquid is also relatively high. The reboiler energy consumption of the MEA solvent is the second highest. $Q_{con}$ and $Q_T$ contribute less. The desorption heat, $Q_{abs}$, contributes the most, and its value is higher than other solvent desorption heats. HEPZ has the lowest reboiler energy consumption, 3.018 GJ/t$CO_2$, which is about 35.2% lower than $Q_{reb}$ of PZ and 11.6% lower than $Q_{reb}$ of MEA, and $Q_{abs}$ is almost the lowest among all solvents, only slightly higher than $Q_{abs}$ of DMEA. The reboiler energy consumption of 1-MPZ, DMEA and AMP is close. The three heat values and their proportions of 1-MPZ and AMP are close.

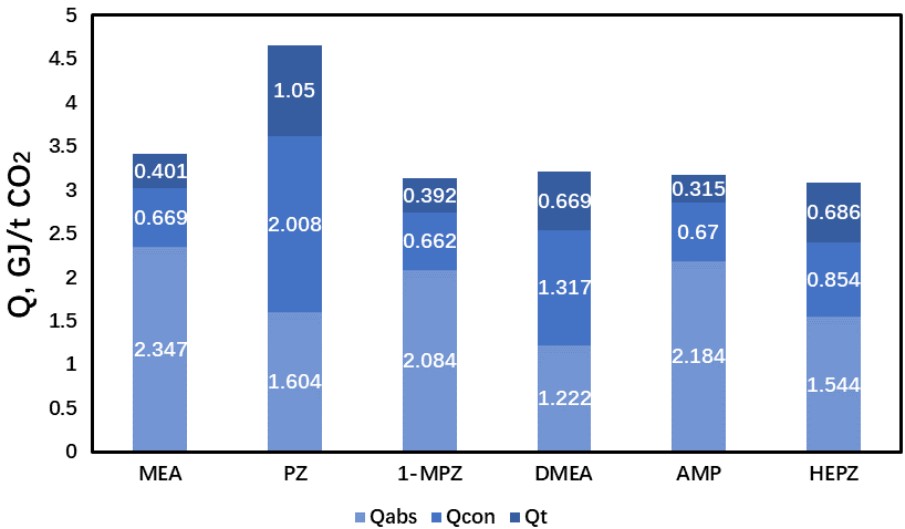

**Figure 9.** The proportion of different heat in the reboiler energy of different solvents.

### 3.3. Comparison of Regeneration Energy, Solvent Loss and Cyclic Capacity of Different Solvents

In addition to regeneration energy, this work also compares the solvent loss and cyclic capacity of each solvent to comprehensively evaluate the economic performance of the solvent. During the operation process, each solvent will have varying degrees of loss as the solution volatilizes in the desorption tower. If the ratio of the loss to the solvent in solution is too large, the concentration of the solution will be diluted. Therefore, it is necessary to constantly replenish the solvent, which increases the cost of the solvent and reduces the economic efficiency of the technology. Therefore, the lower the ratio, the better the

economy of the solvent. It can be seen from Table 3 that the solvent with the lowest ratio is HEPZ, which is 0.02%. The operating temperature of the desorption tower in this process is 123–125 °C. HEPZ has the highest boiling point, so it is the least volatile. 1-MPZ and DMEA have the highest loss ratios because their boiling points are the lowest. In actual operation, the temperature of the desorption tower may be higher. If it exceeds 130 °C, the economic cost for solvent loss will be greater. Therefore, HEPZ has obvious advantages. The cyclic capacity shows the amount of $CO_2$ absorbed by the circulation per kilogram of solution. As a sterically hindered amine, the cyclic capacity of AMP is the highest, which is consistent with the initial theory. The cyclic capacity of HEPZ is 47% lower than AMP, but 64.7% higher than PZ and 21.6% lower than the primary amine MEA. Considering the absorption rate PZ > HEPZ > MEA [33], the economic value of HEPZ as a promotor will be significantly better than MEA and PZ. HEPZ can replace PZ and MEA as an absorbent to configure a mixed amine system with sterically hindered amine AMP. It is expected to be superior to the AMP/PZ system and has value for conducting pilot studies.

**Table 3.** Regeneration energy, solvent loss and cyclic capacity of different solvents.

| Absorbent | $Q_{reb}$ GJ/t$CO_2$ | Cyclic Capacity mol $CO_2$/kg sol | The Solvent Loss mol/s | Loss/RS % |
|-----------|---------|---------------------|----------------|---------|
| MEA | 3.415 | 1.5234 | 0.000342 | 0.14 |
| PZ | 4.657 | 0.7247 | 0.000142 | 0.10 |
| 1-MPZ | 3.136 | 2.0575 | 0.002177 | 0.98 |
| DMEA | 3.207 | 1.0839 | 0.029212 | 3.23 |
| AMP | 3.168 | 2.2883 | 0.002466 | 0.52 |
| HEPZ | 3.018 | 1.1938 | 0.000087 | 0.02 |

## 4. Conclusions

In this work, the influence of process parameters on regeneration energy consumption of post-combustion capture was studied by process modeling on Aspen Plus® platform. Additionally, by comparing the regeneration energy consumption, cyclic capacity and solvent loss during operation of different absorbents, the potential of HEPZ to replace piperazine as the promotor of a mixed amine system was analyzed.

The process parameters that influence regeneration energy consumption mainly include absorbent concentration, liquid to gas (L/G) ratio, $CO_2$-lean loading, $CO_2$ partial pressure of flue gas and the pressure of the desorption tower.

The increase in the concentration of the absorbents will reduce the liquid to gas (L/G) ratio and decrease $Q_{con}$ and $Q_T$. The concentration of different absorbents had different effects on $Q_{abs}$: as the concentration of MEA, 1-MPZ and AMP aqueous solutions increased, $Q_{abs}$ increased, and as the concentration of DMEA and HEPZ increased, $Q_{abs}$ remained almost unchanged. As the absorbent concentration increased, the total energy consumption for regeneration decreased.

The stronger the amine's circulating absorption capacity, the smaller the L/G and the amount of solvent used. The increase of L/G had no effect on $Q_{abs}$: it increased $Q_T$ linearly, and $Q_{con}$ first decreased and then increased. With the increase of L/G, the total energy consumption also decreased first and then increased.

The $CO_2$-lean loading affected the cyclic absorption capacity, which in turn affected the energy consumption. The circulating absorption capacity decreased as the $CO_2$-lean loading increased.

The influence of the $CO_2$ partial pressure of flue gas on regeneration energy is related to the type of solvent. The smaller the slope of the solvent's $CO_2$ solubility curve, the greater the impact of $CO_2$ partial pressure on regeneration energy. When the $CO_2$ partial pressure increased, $Q_{con}$ decreased, and $Q_T$ and $Q_{abs}$ were not affected by $CO_2$ partial pressure. The solubility curve of 40 wt.% MEA had the largest slope, and as the $CO_2$ partial pressure of flue gas increased, the regeneration energy consumption was basically unchanged. The slopes of the solubility curves of 30 wt.% HEPZ and 30 wt.% DMEA

were the smallest, the $CO_2$ partial pressure increased from 5 to 15 kPa and regeneration decreased by about 11% and 20%, respectively.

The change in the pressure of the desorption tower had little effect on energy consumption. The pressure rose by 0.5 bar, the energy consumption was slightly reduced and the temperature of the desorption tower rose by approximately 10 °C. All absorbent systems have a set of process parameters with the largest cyclic absorption and the lowest energy consumption.

With the optimal process parameters, the lowest reboiler energy consumption of HEPZ was 3.018 GJ/t$CO_2$, which was 35.2% lower than that of PZ and 11.6% lower than that of MEA, and the heat of desorption, $Q_{abs}$, was almost the lowest among all absorbents. During operation, HEPZ had the lowest solvent loss. The cyclic capacity was 64.7% higher than PZ and 21.6% lower than the primary amine MEA. The absorption rate was PZ > HEPZ > MEA, and HEPZ has significant economic value as a promotor to replace PZ and MEA.

**Author Contributions:** Conceptualization, S.L. and H.L.; methodology, S.L. and H.L.; software, S.L. and H.L.; validation, S.L. and J.C.; formal analysis, S.L.; investigation, S.L.; resources, J.C.; data curation, S.L. and H.L.; writing—original draft preparation, S.L.; writing—review and editing, J.C.; visualization, S.L.; supervision, J.C. and Y.Y.; project administration, J.C. and Y.Y.; funding acquisition, J.C. All authors have read and agreed to the published version of the manuscript.

**Funding:** This research was funded by the National Natural Science Foundation of China, grant number 21978145, and the National Science and Technology Support Program of China, grant number 2015BAC04B01.

**Institutional Review Board Statement:** Not applicable.

**Informed Consent Statement:** Not applicable.

**Data Availability Statement:** The data presented in this study are available upon request from the corresponding author.

**Conflicts of Interest:** The authors declare no conflict of interest.

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
