# Peer review of "Simulation and Performance Comparison for CO2 Capture by Aqueous Solvents of N-(2-Hydroxyethyl) Piperazine and Another Five Single Amines"

_processes, doi:10.3390/pr9122184_

Round 1

Reviewer 1 Report

Please see red marks and comments in the attachment.

Author Response

Dear Professor,

Thank you very much for the comments.  Please see the attachment for modifications. 

Sincerely,

Jian Chen

Reviewer 2 Report

Li et al presented a parametric study for the evaluation of N-(2-Hydroxyethyl) piperazine (HEP) among other solvents for post-combustion CO2 capture plant. The authors preformed simulations of the post-combustion capture plant in ASPEN in order to obtain the reboiler duty of the solvent as a key performance indicator. The evaluation of the HEP solvent has been done in previous work as outlined by the authors. However, the discussions of the novelty of the current work is lacking. The following comments need to be addressed in the revision:

  1. Please discuss previous parametric studies used for the analysis of post-combustion CO2 capture plant?
  2. The author mentioned “the goal of optimizing process parameters”. Can you please elaborate more on how you have optimized the operating parameters?
  3. Can you please discuss the novelty of this work in comparison to reference 14 cited here?
  4. In table 2, Can you please explain why are using PZ at 10% weight? This will impact the obtained L/G ratio.

Author Response

(The authors gave the same response as above.)
